



# Observational evidence of moistening the lowermost stratosphere via isentropic mixing across the subtropical jet

Jeffery Langille[1], Adam Bourassa[1], Laura Pan[2], Daniel Letros[1], Brian Solheim[1], Doug Degenstein[1],
Daniel Zawada[1]

[1]Institute of Space and Atmospheric Studies, University of Saskatchewan, Saskatoon, S7N 5E2, Canada
[2]National Center for Atmospheric Research, Boulder Colorado, 3090 Center Green Drive, CO 80301, USA

Correspondence to: Jeffery Langille (jeff.langille@usask.ca)

**Abstract.** Isentropic mixing across and above the subtropical jet in the presence of a double tropopause may be a significant mechanism for moistening the lowermost stratosphere. We present an analysis of high spatial resolution two-dimensional measurements of the water vapour distribution that were obtained using the Spatial Heterodyne Observations of Water instrument during a demonstration flight from NASA's high altitude ER-2 airplane. We focus on a set of measurements from 37° to 44° North, obtained on July 21, 2017 during a flight off the West coast of North America, where the instrument sampled the region above and poleward of the subtropical jet. Analysis of these measurements reveals poleward mixing of moist filaments in the region of a double tropopause. These moist filaments are examined in the context of the meteorological fields in the ECMWF (ERA-interim) model output. It is shown that the observed moist filaments are consistent with isentropic mixing across and above the sub-tropical jet. These new observations provide further evidence that the tropospheric intrusion associated with the double tropopause contributes to moistening of the lowermost stratosphere

## 1.    Introduction

The distribution of water vapour in the upper troposphere and lower stratosphere (UTLS) region plays a critical role in the physical processes that couple the region to Earth's climate. This is especially true near the tropopause and in the lower stratosphere where the radiative sensitivity and climate impact of water vapour is the most significant [Solomon et al., 2010]. In this region, mixing between tropospheric and stratospheric air masses results in spatial and temporal variability in the lower stratosphere that is not resolved in current climate models.

Several studies have suggested that this variability provides a mechanism for significant radiative forcing (or feedback) that has the potential to affect long term, as well as, recent climate trends [de Forster and shine, 1999; de Forster and Shine, 2002; Solomon et al., 2010]. Therefore, a detailed understanding of these processes is required in order to fully understand the impact of anthropogenic forcing of the climate [Stohl et al., 2003; Gettelman et al., 2000; Solomon et al., 2010].

The primary pathways for the flux of air mass into the stratosphere are well known [Holton, 1995; Dessler et al., 1999]. The simple picture described in Dessler et al., 1999 is followed here. In the tropics, deep convection on the ascending branch of the Brewer-Dobson circulation [Brewer, 1949; Dobson, 1956] pushes tropospheric air across the tropical tropopause layer (TTL). This air mass slowly ascends into the uppermost stratosphere or overworld (above the 380 K surface) where the isentropic surfaces lie in the stratosphere at all latitudes. During this process, moist tropospheric air is dehydrated as it passes through the cold point in the TTL. Over the course of weeks and months, this dry air is transported to the extratropical lower stratosphere as part of the descending branch of the circulation. The dehydration process in combination with methane oxidation accounts for the low water vapour mixing ratios that are observed in the stratosphere.

In the middle-world, isentropes intersect the tropopause and the tropical upper troposphere and the extratropical lower stratosphere air masses are coupled by isentropic mixing. This region is usually taken to lie between 310 K to 380 K where the layer poleward of the tropopause is referred to as the lowermost stratosphere (Holton et al., 1995). This mixing process is expected to involve finer scale processes and to show filamentary structures (Holton et al., 1995, Appenzeller et al., 1992). Chemical structures of quasi-isentropic mixing have been observed in the vicinity of the subtropical jet in association with Rossby wave breaking events [e.g., Pan et al., 2009; Ungermann et al., 2013].

Stratospheric intrusions during tropospheric fold events result in high stability air pushing deep into the troposphere and subsequently high ozone air (and low water vapour mixing ratios) below the tropopause [Stohl et al., 2003]. On the other hand, during tropospheric intrusions, unstable and moist tropospheric air enters the lower stratosphere. This typically coincides with the formation of a double tropopause above and poleward of the subtropical jet [Pan et al., 2009; Randel et al., 2007a; Randel et al., 2007b; Pan et al., 2010]. In both cases, filamentary structures are observed in chemical species that closely align with these isentropic surfaces. Filamentary structures have also been observed that do not align with these surfaces. Important questions remain regarding the small scale physical mechanisms that control these processes, as well as, which mechanisms dominate moistening of the lower stratosphere.

Current and historical satellite measurements lack the vertical resolution that is required to constrain water vapor processes in the UTLS [Randel and Jensen, 2013] , and merging of individual short term records is challenging, partly due to the limited sampling [Hegglin et al., 2014]. These measurements cannot explain observed variability in lower stratospheric water vapour. Understanding the variability and long term changes in UTLS water vapor at the level required to resolve differences in reanalysis models requires global, high-resolution observations [Randel and Jensen, 2013].

Resolving the spatial structures in the vertical distribution of water vapour requires a vertical resolution of less than several hundred meters and an along track sampling capability on the order of 10 km. Currently, one of the best measurements of this



type are performed using the Microwave Limb Sounder (MLS) on the AURA satellite, which provides 2 km – 5 km vertical resolution in the UTLS with profiles retrieved every ~165 km in latitude [Birner et al., 2006; Müller et al., 2016].

The Spatial Heterodyne Observations of Water (SHOW) instrument is a new limb sounding satellite prototype originally designed and built at York university that is being further developed in collaboration between the University of Saskatchewan
and the Canadian Space Agency to provide high vertical resolution ( < 250 m) measurements of water vapour with high precision (< ±1 ppm) in the UTLS region. The instrument implements a limb imaging spatial heterodyne spectrometer (SHS) to obtain vertically resolved images of the water vapour spectrum using limb-scattered sunlight in a 2 nm spectral window centered on 1364.5 nm (Langille et al., 2017). Each SHOW measurement is inverted using the optimal estimation approach to obtain the vertical water vapour profile for each along-track sample (Langille et al., 2018).


The SHOW prototype flew several demonstration flights on NASA's ER-2 aircraft in July, 2017 in order to validate the measurement approach and to demonstrate the along-track sampling capabilities of the instrument. The SHOW measurement technique, retrieval approach and instrument performance was validated during an Engineering flight that was performed on July 17, 2018 (Langille et al., 2019). Comparison with co-located radiosonde measurements were found to be in excellent
agreement, with differences of < 1 ppm above 15 km (near the thermal tropopause) and < 2-5 ppm below 15 km, due to both natural variability between the observations and measurement precision.

In this paper, we present observations of the two dimensional distribution of UTLS water vapour obtained during a flight on board the ER-2 on July 21, 2017. The flight path was chosen to provide sampling across several degrees of latitude off the
west coast of North America from roughly 34° North to 48° North along the −124.5° West longitude line. We focus our analysis on a one hour window of measurements where SHOW sampled the water vapour distribution near the subtropical jet.

## 2.    The Spatial Heterodyne Observations of Water (SHOW) instrument

The SHOW instrument is spatial heterodyne spectrometer that has been optimized for limb viewing observations of limb-scattered sunlight within a vibrational band of water. The limb is imaged conjugate to the SHS interference fringes such that
each interferogram row and subsequently each spectral row in the image is mapped one-to-one to line of sight at the limb. Each sample provides a vertically resolved spectral image with ~0.03 nm spectral resolution in a 2 nm window centered on 1364.5 nm. These vertically resolved spectral images are inverted using a non-linear optimal estimation approach to obtain the vertical distribution of water vapour. The SHOW measurement technique and retrieval algorithm is discussed in previous publications [Langille et al., 2018; Langille et al., 2019].






| Instrument parameter | Specification |
| --- | --- |
| ER-2 airplane altitude | ~21.34 km (70000 ft  max) |
| Airplane speed | ~760 km/hr (maximum at altitude) |
| Field Of View | 4º vertical by 5.1º horizontal |
| Temporal cadence | 1Hz or 0.5 Hz |
| Spatial sampling at the surface | ~1 km @ 1 Hz |
| Instantaneous angular vertical resolution | 0.0176 degrees |
| Retrieval altitudes | 13 km to 18 km |
| Retrieval grid | 250 m |
| Mass | 222.68 lbs [101 kg] |
| Power | 465 W (peak), 200 W (average) |
| Dimensions | (0.465 m × 1.32 m× 0.38 m ) |
| Spectral Resolution (unapodized) | ~0.03 nm |
| Spectral range | 1363 nm – 1366 nm |

**Table 1 SHOW ER-2 instrument parameters**

The prototype SHOW instrument is optimized for observations from NASA's ER-2 airplane and is mounted in a forward

looking wing pod to observe a 4 degree vertical by 5.1 degree horizontal field of view. Flying at an altitude of 21 km, the

viewing geometry and optical configuration provides a vertical sampling at the limb tangent point of 51 m to 171 m, increasing

towards the ground tangent. Using this configuration, retrievals are performed on a 250 m retrieval grid with no smoothing to

provide an approximate vertical resolution of 250 m from 13 km up to 18 km with precisions better than 1 ppm.  The instrument

can be operated using sampling rates from 0.1 Hz up to 2 Hz mode; however, the measurements discussed in this paper are

obtained using a sampling rate of 1Hz. This provides an approximate raw along track sampling of ~1 km at the surface (or

~0.005 degrees latitude). The primary instrument specifications are listed in Table 1 and the full instrument configuration is

presented in Langille et al., 2019.

### 3.        ER-2 flight path and the metrological background

The measurements discussed in this paper were obtained during a flight on board the ER-2 performed on July 21, 2017 between

18:00 UTC and 19:00 UTC off the Western coast of North America.  For analysis of the meteorological fields within this

measurement window we utilize the ECMWF (ERA-interim) model output. The model data is provided in 1-hour time steps

on a 0.25 degree x 0.25 degree grid (latitude x longitude) at 37 pressure levels from 1 mbar up to 1000 mbar.

The zonal wind at the 175 hPa level (approximately 13 km altitude) for the 18:00:00 UTC time step on July 21, 2017 is shown

in Figure 1.   The zonal wind plot shows the subtropical jet located near 45° degrees North with a secondary branch located

close near 35° degrees North. Both features have jet cores (with winds > 40 m/s) that are located over the Pacific Ocean and

the jets weaken as they approach the coast and become superimposed over North America.



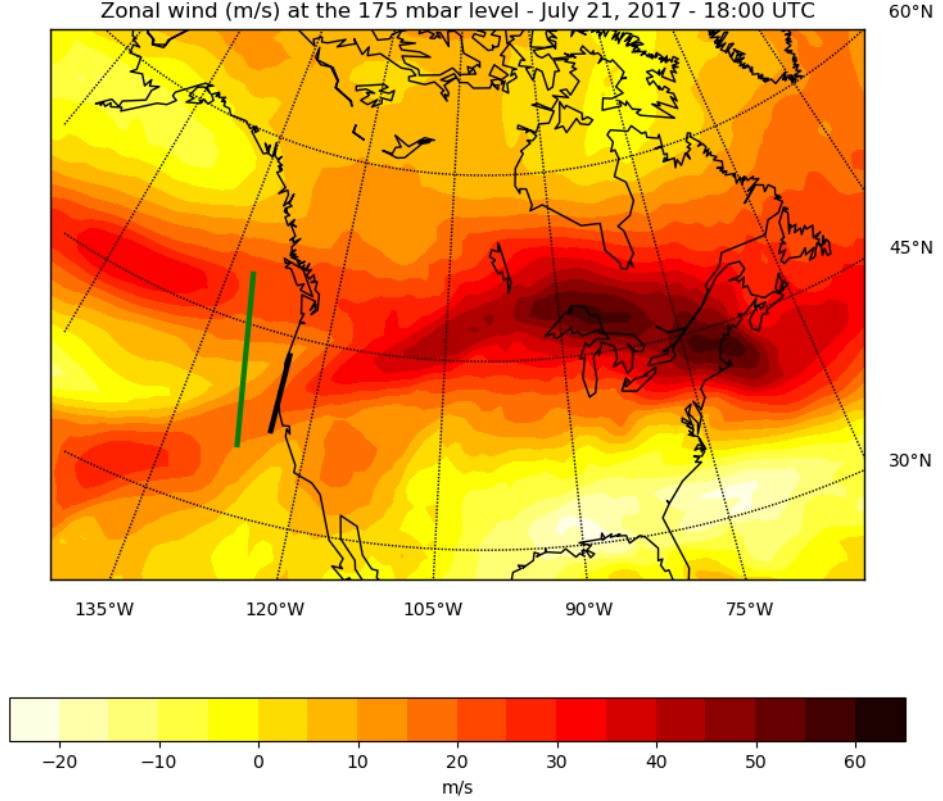

**Figure 1 Zonal wind on the 175 hPa surface. The SHOW measurement track is shown as the dark black line and the closest measurement track of the AURA-MLS instrument is as the dark green.**

The measurement track of the SHOW instrument for the 18:00 UTC to 19:00 UTC time period is shown as the thick black line in Figure 1. Along this track, SHOW obtained high vertical resolution ($< 250$ m) measurements of UTLS water vapour around the tropopause (13 km – 18 km). The sampled region begins near $37°$ N on the southern edge of the primary jet feature and ends on the southern edge of the secondary jet feature near $44°$ N. These measurements were then averaged by latitude to increase the signal to noise ratio, resulting in an along track sampling of approximately 0.32 degrees latitude (approximately 36 km at the ground).

For comparison, the thick green line in Figure 1 shows the measurement track of the AURA-MLS satellite instrument at approximately 21:51 UTC - roughly 2 hours after the SHOW measurements were performed. Along this track, the MLS instrument sampled the same geophysical feature along a slightly different path with an along track sampling of 300 km and a vertical resolution of 3-5 km in the UTLS. The MLS measurements have a courser spatial resolution and the sampling is not exactly coincident with SHOW. Therefore, some differences are expected between the measurements. However, both sensors





sample nearly the same region in the vicinity of the subtropical jet. Therefore, the MLS measurements are used to check for consistency with the meteorological picture in comparison with the SHOW measurements.

## 4.     SHOW Observations

Each of the SHOW limb images was processed to obtain a vertical profile for each sample obtained along the flight track shown in Figure 1. Three example SHOW water vapour profiles are shown in Figure 2 (a-c). The profiles correspond to the

latitude bins centered at 37.4 degrees North, 41.87 degrees North and 43.48 degrees North respectively. Each example shows the set of 10 samples obtained within in each latitude bin (black) and the mean of the sample set (red). The observed variance in the water vapour distribution closely matches the 1-2 ppm measurement error predicted by propagating the noise through the retrieval. The red error bars show the precision for the averaged measurements which is less than < 0.3 ppm for most measurement altitudes.


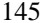

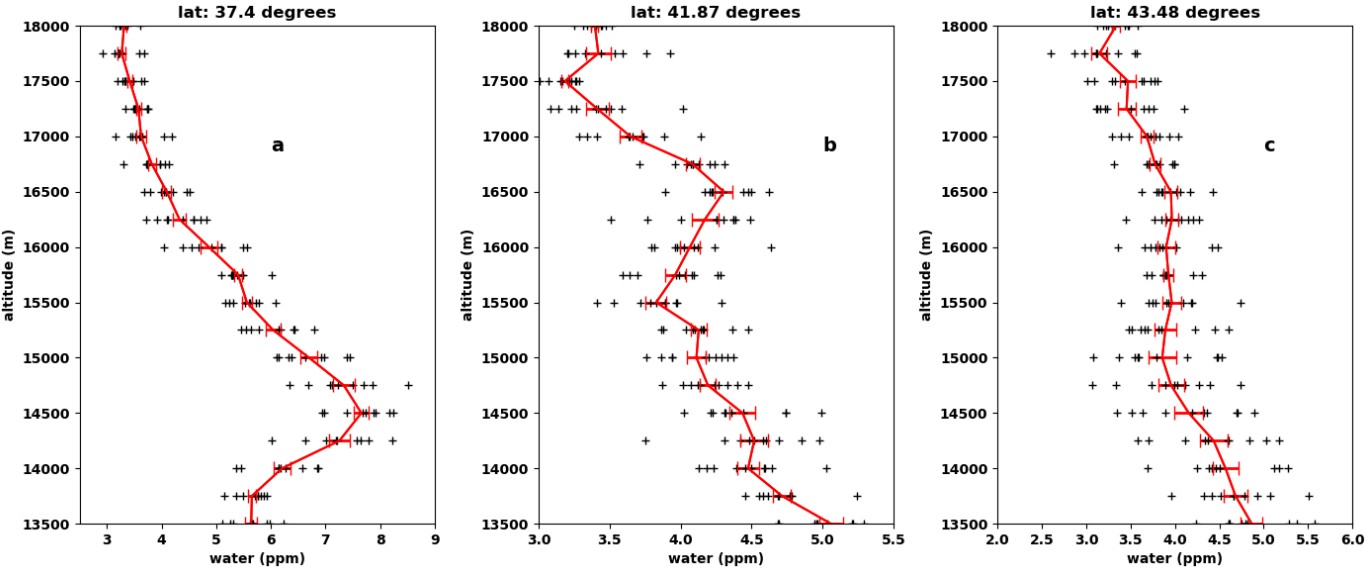

**Figure 2 Example show profiles at 37.4 N (a), 41.87 N (b), and 43.48 N (c). All profiles lie closely along the -124.5 degree longitude line. The black data points correspond to each of the individual profiles and the red line is the line passing through the average of all latitude measurements in each altitude bin. The error bars show the precision of the averaged measurements.**


For the 37.4 N measurement, the water vapour mixing ratio increases to a maximum near 14.5 km and then decreases rapidly with increasing altitude. The water vapour mixing ratio is also found to decrease slightly below 14.6 km. In the current analysis, the lower boundary of the retrieval is at 13.5 km and therefore doesn't capture the expected increase of water vapour at altitudes below 13.5 km.     At 41.87 N a secondary peak in the water vapour profile is observed near 16.5 km. The amount

of water vapour decreases slightly below this peak and then continues to steadily increase with decreasing altitude. Further



along the flight track, at 43.48 N, the peak at 16.5 km has diminished and the amount of water vapour increases slowly with decreasing altitude.

**Figure 3** SHOW measured water vapour profile (from 18:00 UTC to 19:00 UTC) (a) and the potential temperature lapse rate determined from the ECMWF reanalysis for the 19:00 UTC time step along the SHOW measurement track (b). The dark dotted line shows the location of the thermal tropopause. The grey contours show the potential vorticity, several zonal wind contours are shown in light grey, and the light black dotted line shows the 340 K, 380 K and 410 K isentropes respectively.

All of the measured water vapour profiles obtained along the flight track are stacked and plotted as a single data curtain in
Figure 3 (a). The two dimensional profile that is obtained provides a high vertical resolution cross-section of the water vapour
distribution along the flight track. To aid in the interpretation of the observed spatial variability we utilize the dynamical fields
from the ECMWF reanalysis (19:00 UTC time step) and plot several zonal wind (black) and potential vorticity (PV) (orange)
contours on top of the water vapour measurements. The potential temperature lapse rate determined from the reanalysis data

($PTLR = \Delta\theta/\Delta z$) is shown in Figure 3 (b) for the same region and time step. In both figures, the 340 K, 380 K and 410 K
isentropes are shown as the thin dotted lines and the thick black dots identify the location of the thermal tropopause.

The subtropical jet is in Figure 3 centered at the tight zonal wind contours (light grey) near 39.9 degrees at an altitude between
11-12 km. South of 39.9 degrees latitude, the thermal tropopause sits at an altitude of close to 15.5 km. Near 39.9 degrees we

observe a break in the tropopause and record a double tropopause that extends from 39.9 degrees to 42 degrees. At the break,
the primary tropopause drops to below 13.5 km and remains at this altitude or lower for higher latitudes. The tropopause break
is a common feature of the thermal tropopause definition and is a ubiquitous feature found on the poleward side of the
subtropical jets in observations [Randel et al, 2007 b], as well as, climate models [Manney, 2014].

From Figure 3, the 410 K isentrope lies entirely in the stratosphere (in the overworld) at all latitudes. Above the 410 K
isentrope, the water vapour mixing ratio is observed to have values between 3.0 ppm – 3.8 ppm which defines the background
water vapour mixing ratio in the lowermost stratosphere. Near the tropopause (in the middleworld), sharp spatial structures are
resolved that have gradients on the order of 0.5 ppm per 250 m sampling bin. SHOW does not record the water vapour
distribution below the 340 K isentrope since the retrieval cuts off at an altitude of 13.5 km.

Following from Figure 3 (a), SHOW recorded a large peak in the water vapour mixing ratio at roughly 14.6 km that extends
from around 37° N to 40.5° N. The location of the thermal tropopause tracks the strongest gradients in the water vapour profile
since the layer of air just above the tropopause acts as a strong barrier to vertical motions, resulting in a rapid decrease in water
vapour above the tropopause. Correspondingly, a sudden drop in the amount of water vapour is recorded near 39.9° N

coinciding with the tropopause break. After the break, the strongest gradient in the water vapour profile drops to nearly 13.5
km and continues to closely track the thermal tropopause.

Most interestingly, a well-defined moist filamentary structure is observed above and poleward of the tropopause break that
coincides with the presence of a double tropopause. This structure has a water vapour mixing ratio that is larger than the

background stratosphere by 1-1.8 ppm from 39.5° − 42°North within a 1 km region that is centered near ~16.5 km. The
structure also roughly aligns with the isentropic surfaces.

It is widely accepted [Pan et al., 2009; Randel et al., 2007b; Gettelman et al., 2011] that the formation of the double tropopause on the poleward side of the subtropical jet is the result of the intrusion of low static stability air from the subtropical troposphere into the mid-latitude lowermost stratosphere. Such intrusions result in isentropic mixing across and above the subtropical jet
[Gettelman et al., 2011]; therefore, UTLS species, such as water vapour and ozone act as tracers of these events. The rough alignment of the isentropic surfaces with the moist filament recorded by SHOW suggests isentropic mixing between tropospheric air and the lowermost stratosphere.

To support this interpretation we examine the PV and PTLR in the reanalysis data. Accordingly, the dark grey contours in
Figure 3 (a) show the potential vorticity from 2 PVU to 14 PVU in 2 PVU increments. The surface defined by the 6 PV contour appears to characterize a dynamical surface that separates tropospheric and stratospheric air. Note that a patch of low PV air (< 4 PVU) is recorded near 15 km with a spatial range that extends from $39.5° - 42°$ N. The shape of the feature is strikingly similar in structure to the moist filament observed by SHOW at roughly 1.5 km higher altitude (~16.5 km).

The associated PTLR (shown in Figure 3 (b)) also tracks the overall spatial structure that is observed in the water vapour distribution. This is anticipated since the PTLR provides a measure of the static stability of the air. Therefore, sharp gradients are expected to mark the separation between low stability tropospheric air and high stability stratospheric air. Along the SHOW measurement track, tropospheric air is primarily characterized with a PTLR < 12 K/km and stratospheric air is characterized with a PTLR > 12 K/km. A filamentary structure with PTLR < 12 K/km is observed to extend from $39.5° - 42°$ near ~15 km.
This structure of low static stability air coincides with the presence of the double tropopause and matches a similar feature that is evident in the potential vorticity, as well as, the moist filamentary structure observed by SHOW at an altitude of ~16.5 km.

The offset in altitude between the filamentary structure observed by SHOW and the spatial structure that is observed in the PV and PTLR fields could be due to several reasons. For example, the SHOW instrument is a limb viewing instrument; therefore,
this may be a consequence of the limb viewing geometry. In addition, the meteorological data is determined from a model and it is entirely possible that the model does not fully capture the true geophysical state of the atmosphere. We will see through a comparison with the MLS measurements of water vapour and ozone that a similar shift is also observed in that dataset. Therefore, it is most likely that the reanalysis does fully represent the finer scale structure during this event.

## 5.    AURA MLS ozone and water vapour

The AURA-MLS measurements of water vapor and ozone that were obtained along a nearly coincident measurement track are shown in Figure 4 (a) and Figure 4 (b) respectively. The corresponding PTLR plot is shown in Figure 4 (c). For this comparison we use the 22:00 UTC time step of the ECMWF reanalysis since it is the closest time step to the MLS measurements which occurred at close to 21:50 UTC.

The distributions of the two trace species have a spatial structure that matches the general shape of the structure observed in the PTLR plot and PV contours. As expected, the vertical distributions of the trace species are basically inverted, with water vapour decreasing with increasing altitude and vice versa for ozone. Most importantly, a filamentary structure is observed that extends from 36 N to 42 N near 16 km and coincides with the presence of a double tropopause. Again, the feature matches a similar structure that is observed in the corresponding PTLR plot and PV contours at a lower altitude (~ 15 km).

Taking the sharpest gradient in the PTLR to define the boundary between tropospheric and stratospheric air we see that tropospheric air is primarily characterized with a PTLR < 12 K/km and stratospheric air is characterized with a PTLR > 12 K/km. Therefore, as was the case with the SHOW measurements, the observed filamentary structure with PTLR < 12 K/km is consistent with the intrusion of a low static stability air from the subtropical troposphere into the mid-latitude lower stratosphere. This results in the formation of the double tropopause and the transport of moist tropospheric air and ozone

depleted air into the mid-latitude lower stratosphere.

The spatial structures recorded by SHOW (Figure 3) and MLS (Figure 4) during this intrusion event are strikingly similar and are consistent with spatial structures in the meteorological fields. A direct comparison shows that both instruments recorded similar amounts of water vapour in the vicinity of the subtropical jet. They both capture the moist filament near 16 km, as well

as, the dry regions near 13.5 km in the lower latitude portion of the measurement tracks. The measurements are not expected to have exact agreement since the MLS measurements are made along a flight track that samples a slightly different region of the atmosphere. Also, the limb viewing geometry from a satellite is different from the aircraft and the MLS measurements have a lower vertical resolution (3-5 km) compared to the SHOW measurements (250 m). These differences introduce additional biases between the measurements that are difficult to quantify without an exact coincidence where retrieval

averaging kernels could be used to compare more quantitatively although degrading of the SHOW measurements to MLS resolution is better left to a separate validation exercise.

Interestingly, the spatial structures observed in the MLS ozone and water vapour profiles are both shifted to a higher altitude relative to the PTLR and PV structures. The consistency of the shift between SHOW and MLS suggests a slight bias in the

ECMWF reanalysis data; although, it is unclear exactly the cause of the bias. Regardless, it is clear that the spatial variability observed in the MLS ozone and water vapour measurements, in light of the higher resolution SHOW observations, is consistent with isentropic mixing and a tropospheric intrusion event associated with a double tropopause.






**Figure 4** MLS measured water vapour profile (a), ozone (b) and the potential temperature lapse rate determined from the ECMWF reanalysis for the 22:00 UTC time step along the MLS measurement track (c).



## 6. Spatial extent of the event

To complete the picture it is prudent to examine the spatial extent of the suspected intrusion event. This is achieved by examining the PTLR at the 22:00 UTC time step in Figure 5(a) and Figure 5(b) for a longitude of 235.5 degrees and 230 degrees respectively. Latitudes from 30 degrees to 70 degrees North are shown to give a view of the larger dynamical picture.

As before, the black contours show the zonal wind, the thick black dotted line shows the location of the thermal tropopause and the light black dotted lines show the 350 K and 400 K isentropes.

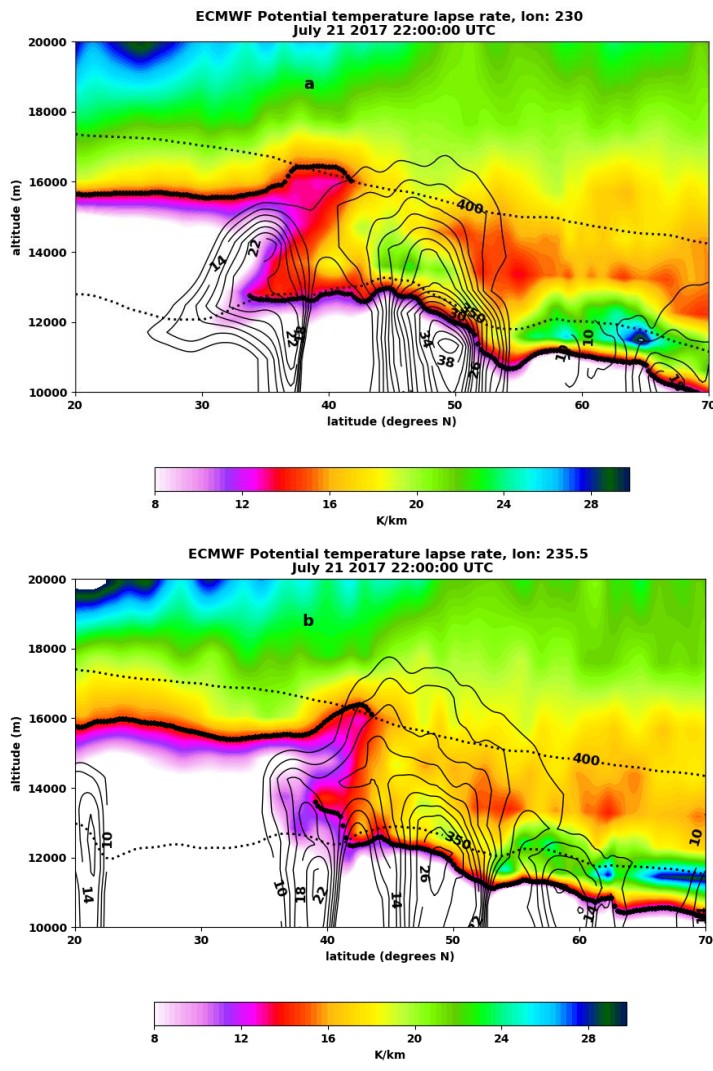


**Figure 5 ECMWF calculated potential temperature lapse rate along the 235.5 and 230 degree longitude line determined from the ECMWF meteorological data at the 22:00 UTC time step**



Along the 235.5 degree longitude (near the SHOW flight track) we see that the thermal tropopause is located close to 16 km below 39 degrees. Above 39 degrees, a double tropopause forms that extends from 39 degrees North to 44 degrees North. On the other hand, for the 230 degree longitude (near the MLS flight track), the thermal tropopause sits close to 16 km below 33 degrees and the double tropopause extends from 32 degrees to 42 degrees. In both cases, the presence of the double tropopause coincides with a laminar structure that has PTLR < 12 K/km air that extends above and poleward of the jet. Only marginal differences are observed between the 19:00 UTC (Figure 3 (b)) and the 22:00 UTC time steps in the vicinity of the suspected intrusion.

The PTLR in the larger geographical region shows that the suspected intrusion and the associated double tropopause has a spatial extent that extends between 5-10 degrees poleward of the subtropical jet over the Pacific Ocean. The spatial structures observed in the measurements of water vapour with SHOW and ozone and water vapour with MLS act as tracers of this intrusion. In the SHOW measurements, the intrusion extends from 39.5 degrees to 43 degrees, whereas, in the MLS data, the intrusion extends over more than 10 degrees latitude.

## 7.    Summary and Conclusion

In this paper we presented two-dimensional measurements of the water vapour distribution above and poleward of the subtropical jet. These measurements were obtained using the newly developed SHOW instrument during a sub-orbital demonstration flight on board NASA'S high altitude ER-2 airplane on July 21, 2017. The high spatial resolution sampling provided by SHOW revealed a moist filament that coincided with a double tropopause on the poleward side of the subtropical jet. Nearly coincident measurements of water vapour and ozone obtained using the AURA-MLS instrument recorded spatial structures that were consistent with the SHOW observations. However, it was shown that the vertical resolution provided by SHOW reveals fine spatial structure that is not revealed by the MLS measurements and is not captured in the metrological fields of the reanalysis model output. Therefore, the SHOW measurements provide additional insight into the mechanisms that are responsible for the observed mixing between the air masses.

Analysis of the water vapour distribution indicates that the observed moist filament is the result of isentropic mixing across and above the subtropical jet. The observed variability suggests a complicated mixing layer in the vicinity of the subtropical jet and supports the suggestion that tropospheric intrusions in the vicinity of a double tropopause are a potentially significant mechanism responsible for moistening the lower stratosphere.



310

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
