# Peer review of "Observational evidence of moistening the lowermost stratosphere via isentropic mixing across the subtropical jet"

_Atmospheric Chemistry and Physics, 2019_

## Referee Comment (RC1) · Anonymous Referee #1 · 6 Jan 2020

Observational evidence of moistening the lowermost stratosphere via isentropic mixing across the subtropical jet J. Langille et al.

The authors present results from a new remote sensing instrument designed for satellite-borne high-vertical resolution limb soundings o f water vapor in the upper troposphere/lower stratosphere. The instrument was mounted on board a high-altitude research aircraft and a single cross section obtained of an intrusion of tropospheric air into the lowermost stratosphere, just above the substropical jet is presented. The cross section provides evidence of a moist filament of tropospheric air being mixed poleward into the lower most stratosphere.

[Figure]

Such events are of physical and climatic interest given the role of these events in moistening the extratropical lower stratosphere and thus determining water vapor concentrations in a region important for climate forcing. The observations reveal features at very fine vertical length scales (< 1 km) which are difficult to observe and to model, although the MLS observations do seem to capture the filament in question to some extent. This filament of elevated water vapor coincides with the (upper) tropopause near the 400 K isentropic surface. The authors suggest tentatively that there may be a bias in the ECMWF reanalysis in the form of a vertical offset of the dynamical fields, in that there is a similarly located region of low potential vorticity air somewhat lower in altitude.

Regardless of this suggestion, this observation of a fine filamentary structure extending into the lowermost stratosphere is certainly worthy of publication, but since the scope of the paper is, as set out by the authors, primarily to present the scientific value in these test observations (not simply to validate them) I think it is appropriate to ask for a bit more follow up analysis on some of the details.

Firstly, the newer ERA 5 reanalysis should be used instead of ERA Interim. It has substantially higher vertical and horizontal resolution, and the data is easily available. It would be also worth looking at model level data which is much finer than the data provided on the 37 pressure-level grid. I am not convinced of the inferred vertical (see below), but in any case, the comparison would be much more relevant in the context of the more modern product.

Secondly, emphasis has been placed on the vertical and meridional structure of this filament, but the only synoptic details we are given is in the form of an isobaric wind map at 175 hPa. It could be very enlightening to see some maps of potential vorticity on the 400 K and the 380 K isentropic surfaces with a domain comparable to Figure 1, in order to distinguish the filament from the layer of air between the double tropopause structure highlighted in Fig. 3a. These may have quite different horizontal structures that could shed light on the fine vertical structure of the observed water vapor. On a

related note, I don't think the text includes a discussion of the line-of-sight resolution of the measurements (i.e. in the longitudinal direction in the current geometry).

With regards to the suspected offset in the reanalysis output, this is certainly a difficult region to capture correctly and so it seems plausible to me that such an offset could exist. However, it's also possible that the water vapor transport is not aligned with the lowest PV anomaly, or that the layer of most effective intrusion is not where the PV gradients are strongest (after all the meridional PV gradients act as a horizontal mixing barrier that discourages such intrusions). The requested figures should provides some clarity on this point. A related dynamical point is that the potential temperature lapse rate is not a materially conserved quantity, while PV is (up to diabatic processes); this point should probably be stressed more clearly in the text.

One final minor comment: on line 168 reference is made to orange contours in Figure 3 that I think are in fact dark gray; the orange contours only show up in Figure 4.

---

## Referee Comment (RC2) · Anonymous Referee #2 · 10 Jan 2020

In this manuscript, the authors present the analysis of a double tropopause-intrusion event as a case study to validate the new SHOW instrument. The authors compare their results to reanalysis and satellite data.

I find the work here presented exciting, and no doubt, it lets to get some insight on the potential of the instrument. I want to congratulate the authors for the work developed. At the same time, I have to say that I have detected several mistakes along with the manuscript and that I think that both the analysis and the presentation can be and should be improved. I am familiar with the topic here discussed, and I have found the description sometimes confusing and incomplete. Therefore, those not so familiar with

the matter could find it challenging to understand some issues.

One of the more puzzling issues that I have found in the manuscript is that the Introduction is poor in number and the use of appropriate references. There are some striking examples along with the paper because they involve some of the coauthors. This lack of adequate references makes the discussion about the case study not well balanced and can complicate the reader to have a general perspective of the phenomenon. Below I address this issue with some suggestions where it corresponds.

For example, in the first paragraph of the Introduction, it would be appropriate to cite a work that supports the statement on the limitation of models. Sections 6.2.4 and 6.3 of Gettelman et al. (2010) deal with it (https://doi.org/10.1029/2009JD013638). Also, it is usual to cite Gettelman and Forster (2002) when you refer to the CPT (line 37)(https://doi.org/10.2151/jmsj.80.911). The physical mechanisms mentioned (line 54) are well explained with a model and radiosonde data by Ferreira et al. (2015) (https://doi.org/10.1002/qj.2697). It provides an excellent discussion of some of the most relevant constraints, and it would be worthy of citing it to let the reader get some insight on them.

In lines 57-61, the authors discuss the limitations of satellite data. Indeed, they use AURA-MLS for comparison purposes here. I think that they should cite the works validating WV profiles of AURA-MLS for the SPARC Data Initiative, as they provide the background on the validity and limitations of the measurements. At least one of the coauthors of this manuscript is also coauthor of such works:

Toohey et al. (2013) https://doi.org/10.1002/jgrd.50874 Hegglin and Tegtmeier (2017) The SPARC Data Initiative: Assessment of stratospheric trace gas and aerosol climatologies from satellite limb sounders. SPARC Report No. 8, WCRP-05/2017.

Related to the Introduction: the manuscript has two parts, the validation of the instrument and the case study. Therefore, I think that all the information relevant for the case study that lets to interpret the results should be presented first, included in section 1.

[Figure]

In this vein, the current section 6 should be moved earlier in the manuscript, before beginning the analysis and interpretation of the results. Also, the current figure 1 is right; still, I think that it would be good to include a similar isobaric synoptic map (to check the meteorological situation) and the corresponding map for the first lapse rate tropopause. Doing it would let the reader have a broad picture of the situation. Double tropopauses can happen because of several different conditions, and a priori all of them should be had into account. To do it, all this information is relevant. After it, I suggest to include a brief sentence discussing how the region chosen for the ER-2 flight is one of the central global areas of occurrence of double tropopauses including summer as Añel et al. (2008) shows (https://doi.org/10.1029/2007JD009697). The other work cited in the text and typical about the study of double tropopauses, Randel et al. (2007) do not show them for July over the region studied in this work.

In Figure 2, I had to realise that the values of the horizontal axis are different for each subplot. Right now, it is harder to visualise the latitudinal variation and the assessment of the vertical 'peaks', but it is necessary to be able to compare all of them adequately. Therefore, please, use the same axis for every subplot. Also, it would be helpful to contextualise the air masses if you can add a horizontal line at the level of the thermal tropopause (first and second, if possible). In the caption, you have missed the degree symbol before the cardinal points.

Regarding Figure 3, I have several concerns that should be clarified and better discussed:

First of all, it would be useful in you can include the longitude value in the caption. Secondly, the authors do not say how they have computed the thermal tropopause. Did they use its definition (WMO, 1957)?. Was it retrieved from reanalysis?. It has to be clarified. Also, the use of model levels for the reanalysis could have improved the discussion. If you can use them, it would be better.

About the plots: in Fig. 3a the isentropic entrainment in the lowermost stratosphere

reaches 40.5 degrees N. However, in this region, the PV values are large (up to 6 PVUs, at least 5 PVUs). No doubt, the WV is of tropospheric origin, but such PV values are much higher than acceptable for tropospheric air. At these latitudes, the larger values expected for tropospheric air are 3.5-4 PVUs. If you check your Fig. 4a it seems clear that the 6 PVUs value that you mention in the text as a value for the tropospheric air, is seen in AURA, not so much in the SHOW measurements (and the AURA measurement fits better with the shape of the potential temperature lapse rate in Fig. 3b). Wang and Polvani (2011) (https://doi.org/10.1029/2010JD015118) and Añel et al. (2012) (http://dx.doi.org/10.1100/2012/191028) have already shown with idealized experiments and lagrangian models how the equatorward movement of air masses trough tropopause breaks at midlatitudes is also very important (and indeed they did it for regions close to the one studied here). Checking the Figure 4b, it could be argued that there is a fingerprint of the movement of stratospheric air equatorward through the break, because of the higher values of ozone that reach the 4 PVUs (near to the more accepted tropopause value) and 36.5 degrees N. Therefore, I think that you need to write the paragraphs from line 204-216 with a more complete discussion and better balance.

A right way of checking the reality of the movement would be with using a lagrangian transport model (as in Añel et al. 2012). If you can include it, it would be a great addition to the manuscript, but I realise that it is not the goal of this work, so I do not consider it a 'must' here. But given that you do not provide it and on the ground of the tasks that I mention above, along with the text, you should relax the language and the level of the statement about poleward isentropic mixing. For example, in line 197, where you say 'it is widely accepted' because there are many different synoptic situations.

Finally, the 'summary and conclusion' section is short. I would include some discussion on the error of SHOW and how it could have impacted the results here presented. Also, I would find it interesting to include a reflection on the limitations of SHOW to sample similar episodes in more poleward latitudes. Given that SHOW seems to have

a restriction below 13.5 km and that poleward the tropopause height decreases, is SHOW limited to sample these episodes only around the subtropical jet?. In line 304, the terminology of 'tropospheric intrusions' is used again. As said before, I think that talking about 'tropopause breaks' is correct in the context of this work.

References: - The list of references is in the incorrect order. - Randel et al. 2007a and 2007b are not distinguished in the list of references.

Line 28 - de Forster and Shine Line 30 - Gettelman and Sobel, 2000 Line 44 - Appenzeller and Davies, 1992 Line 52 - Pan et al. 2010 is not listed among the references Table 1 - units of speed – km/h (not km/hr) Line 112 - the international units of pressure are hPa, not mb Line 133 - coarser Line 168 - there is not an orange line in the figure Lines 176-178 - this sentence is redundant. It has been discussed earlier in the text. I suggest removing it. Lines 187-189 - in line 188 when you refer to the tropopause, it is hard to know if you refer to the first or the second; please clarify it. Also, it would be useful in you can include in the plot something (an 'A' and a 'B,' a star and a square,...) to make clearer to what region you refer. Lines 248-251: the last sentence is obvious and can undermine the achievements of the SHOW instrument. I recommend to move it to the conclusions as a final reflection. Line 269 - 20 degrees Figure 5 - caption - I would say '230 and 235.5'. I read the plots from top to bottom and the one on the top corresponds to 230. As it is now, it can be not very clear.
* * *

---

## Author Comment (AC1) · 6 Mar 2020

**Overall Author Response:**

Thank you very much for your comments and suggestions regarding our manuscript. We agree with most your suggested changes and have included several edits to the final manuscript that reflect these changes.  We believe that the associated changes and additional analysis provides better context for the observations and makes for a more complete paper. We have responded directly to your comments below in red (A.1.#) and have identified where the corresponding changes have been made in the manuscript.  In addition to these changes, there have been several format and structural changes that were made to the manuscript in response to suggestions and comments from the second reviewer. Please refer to the responses to the second reviewer for a description of those changes.  Note that several new Figures have been added to the manuscript in order to expand the analysis.  We have also edited the figures to have the same colormap throughout the paper.

Summary of key revisions made to the paper:

1) A synoptic scale meteorological analysis is included for the Rossby wave breaking event that resulted the observed dynamical structure.
2) Discussion of the process-consistency despite the specific differences between SHOW water vapor structure and the ERA5 dynamical field is made to clarify that multiple factors can contribute to the specific differences, including the physical factor that when wave breaking result in irreversible mixing, the air mass composition loses its correlation with PV as a dynamical tracer.
3) More focused in the objectives and take-home messages of this paper to present the new observational evidence of water vapor transport into lowermost stratosphere driving by Rossby wave breaking and instrument capability and potential impact on stratospheric water vapor budget. Eliminated the additional discussions on the scale of the event and further dynamical analysis to avoid distracting from the main messages.
4) The abstract has also been edited accordingly

**Anonymous Referee #1

The authors present results from a new remote sensing instrument designed for satellite-borne high-vertical resolution limb soundings of water vapor in the upper troposphere/lower stratosphere. The instrument was mounted on board a high-altitude research aircraft and a single cross section obtained of an intrusion of tropospheric air into the lower most stratosphere, just above the subtropical jet is presented. The cross section provides evidence of a moist filament of tropospheric air being mixed poleward into the lower most stratosphere.

Such events are of physical and climatic interest given the role of these events in moistening the extratropical lower stratosphere and thus determining water vapor concentrations in a region important for climate forcing. The observations reveal features at very fine vertical length scales (< 1 km) which are

difficult to observe and to model, although the MLS observations do seem to capture the filament in question to some extent. This filament of elevated water vapor coincides with the (upper) tropopause near the 400 K isentropic surface. The authors suggest tentatively that there may be a bias in the ECMWF reanalysis in the form of a vertical offset of the dynamical fields, in that there is a similarly located region of low potential vorticity air somewhat lower in altitude.

Regardless of this suggestion, this observation of a fine filamentary structure extending into the lowermost stratosphere is certainly worthy of publication, but since the scope of the paper is, as set out by the authors, primarily to present the scientific value in these test observations (not simply to validate them) I think it is appropriate to ask for a bit more follow up analysis on some of the details.

**Author Response (A.1.1)**

We agree with your request to present some follow up analysis that serves to highlight the scientific value of the SHOW measurements.  Therefore, we have added several new Figures (Figure 1- 4) and further analysis of the meteorological fields that provides evidence that the observed moist filament is due to isentropic mixing following a Rossby wave breaking event in the days preceding the ER-2 flight. This is discussed in more detail in A.1.3 below.   We have also expanded the discussion in section 6 to examine potential errors in the SHOW measurements, the determination of the upper and lower boundary of the retrieval and reasons for potential biases between SHOW, MLS and the reanalysis data. It also emphasizes the spatial structures in the PTLR in the context of the new analysis of the meteorological fields in support of the suggestion that the moist filament is of tropospheric origin (tropospheric intrusion).

Firstly, the newer ERA 5 reanalysis should be used instead of ERA Interim. It has substantially higher vertical and horizontal resolution, and the data is easily available. It would be also worth looking at model level data which is much finer than the data provided on the 37 pressure-level grid. I am not convinced of the inferred vertical (see below), but in any case, the comparison would be much more relevant in the context of the more modern product.

**Author Response (A.1.2)**

 We are actually using ERA 5, not ERA-interim. The data is provided in 1-hour time steps on a 0.25-degree x 0.25-degree grid.   However, the ERA5 products available to us, unfortunately, still has 37 level, with corresponds to 25 hPa vertical resolution in the tropopause region. This is much coarser than the measurement.

Secondly, emphasis has been placed on the vertical and meridional structure of this filament, but the only synoptic details we are given is in the form of an isobaric wind map at 175 hPa. It could be very enlightening to see some maps of potential vorticity on the 400 K and the 380 K isentropic surfaces with a domain comparable to Figure 1, in order to distinguish the filament from the layer of air between the double tropopause structures highlighted in Fig. 3a. These may have quite different horizontal structures that could shed light on the fine vertical structure of the observed water vapor. On a related note, I don't think the text includes a discussion of the line-of-sight resolution of the measurements (i.e. in the longitudinal direction in the current geometry).

**Author Response (A.1.3)**

In order to address your first point, we have included several additional Figures and associated analysis that we believe provides the necessary context for the case study that is presented in the paper. We have added several paragraphs to Section 3 that discusses these Figures.

Specific changes:

1.  We have included a new figure (Figure 2) showing 3 - 48-hour time steps (each day at 20:00 UTC) of PV on the 380 K surface for the 6 days leading up to the date of the case study. The Figure clearly shows a Rossby wave-breaking event has occurred in the days preceding the flight that results in mixing along the subtropical jet.

2.  We have added a new figure (Figure 3) shows the PV on the 380 K (Figure 3 (a)) and 400 K (Figure 3 (b)) surfaces for the 07/21/2017 18:00 UTC time step. In the Figures, the tropospheric and stratospheric air masses are separated by the 6PVU contour on the 380 K surface and 8 pvu on the 400 K surface.  Here it is observed that the mixing associated with the Rossby wave breaking results in a long low PV "tongue" consistent with tropospheric air that extends from the Western Pacific and tracks the subtropical jet across North America.

3.  To characterize the vertical structure we have included a Figure (now Figure 4) that shows the height of the thermal tropopause and the location/extent and height of the secondary tropopause for the 07/21/2017 18:00 UTC time step. In these figures one can clearly see that there are several double tropopause regions located on the poleward of the subtropical jet. The SHOW measurements track crosses one of these regions.  While additional time steps are not shown, it is useful to point out that the regions of double tropopause vary in extent from time-step to time-step. In fact, the double tropopause region that SHOW crosses becomes larger near the 21:00 UTC time step.  A paragraph has been included in the text that discusses this Figure. We believe that the updated analysis provides the relevant context for the case study and justifies the suggestion that the moist filament observed along the second tropopause in Figure 6 (a) is likely of tropospheric origin.

4.  Regarding the line of sight resolution in the longitudinal direction, the SHOW instrument averages over 4 degrees in the horizontal by making use of anamorphic input optics. Therefore, no horizontal scene information is obtained. A sentence has been added after the second sentence of the second paragraph in Section 2 to clarify this point for the reader.

With regards to the suspected offset in the reanalysis output, this is certainly a difficult region to capture correctly and so it seems plausible to me that such an offset could exist. However, it's also possible that the water vapor transport is not aligned with the lowest PV anomaly, or that the layer of most effective intrusion is not where the PV gradients are strongest (after all the meridional PV gradients act as a horizontal mixing barrier that discourages such intrusions). The requested figures should provides some clarity on this point. A related dynamical point is that the potential temperature lapse rate is not a

materially conserved quantity, while PV is (up to diabatic processes); this point should probably be stressed more clearly in the text.

**Author Response (A.1.4)**

The text has been updated to be clearer on this point.  Specifically, the lines highlighting a potential bias between the reanalysis and observations have been removed; however, the identification of the misalignment between the two is discussed.  Several new paragraphs have been added to Section 5 that examine the PV and layered thermal structure in the context of mixing following the Rossby wave breaking.  It is clarified that it is physically possible (and reasonable) that the dynamical field and chemical structure are no longer intact, which is a sign of an irreversible transport. In addition, the ERA5 products are given at a much coarser resolution than the SHOW measurements. We include a paragraph in the discussion of Section 6 that clarifies these points.

One final minor comment: on line 168 reference is made to orange contours in Figure 3 that I think are in fact dark gray; the orange contours only show up in Figure 4.

**Author Response (A.1.5)**

The text has been modified so the appropriate "grey" contour is mentioned.

---

## Author Comment (AC2) · 6 Mar 2020

**Overall Author Response:**

Thank you very much for your comments and suggestions regarding our manuscript. We agree with most your suggested changes and have included several edits to the final manuscript that reflect these changes. We believe that the associated changes and additional analysis provides better context for the observations and makes for a more complete paper. We have responded directly to your comments below in red (A.2.#) and have identified where the corresponding changes have been made in the manuscript. In addition to these changes, there have been several format and structural changes that were made to the manuscript in response to suggestions and comments from the first reviewer. Please refer to the responses to the first reviewer for a description of those changes. Note that several new Figures have been added to the manuscript in order to expand the analysis. We have also edited the figures to have the same colormap throughout the paper.

Summary of key revisions made to the paper:

1) A synoptic scale meteorological analysis is included for the Rossby wave breaking event that resulted the observed dynamical structure.
2) Discussion of the process-consistency despite the specific differences between SHOW water vapor structure and the ERA5 dynamical field is made to clarify that multiple factors can contribute to the specific differences, including the physical factor that when wave breaking result in irreversible mixing, the air mass composition loses its correlation with PV as a dynamical tracer.
3) More focused in the objectives and take-home messages of this paper to present the new observational evidence of water vapor transport into lowermost stratosphere driving by Rossby wave breaking and instrument capability and potential impact on stratospheric water vapor budget. Eliminated the additional discussions on the scale of the event and further dynamical analysis to avoid distracting from the main messages.
4) The abstract has also been edited accordingly

In this manuscript, the authors present the analysis of a double tropopause-intrusion event as a case study to validate the new SHOW instrument. The authors compare their results to reanalysis and satellite data. I find the work here presented exciting, and no doubt, it lets to get some insight on the potential of the instrument. I want to congratulate the authors for the work developed. At the same time, I have to say that I have detected several mistakes along with the manuscript and that I think that both the analysis and the presentation can be and should be improved. I am familiar with the topic here discussed, and I have found the description sometimes confusing and incomplete. Therefore, those not so familiar with the matter could find it challenging to understand some issues.

One of the more puzzling issues that I have found in the manuscript is that the Introduction is poor in number and the use of appropriate references. There are some striking examples along with the paper

because they involve some of the coauthors. This lack of adequate references makes the discussion about the case study not well balanced and can complicate the reader to have a general perspective of the phenomenon. Below I address this issue with some suggestions where it corresponds.

For example, in the first paragraph of the Introduction, it would be appropriate to cite a work that supports the statement on the limitation of models. Sections 6.2.4 and 6.3 of Gettelman et al. (2010) deal with it (https://doi.org/10.1029/2009JD013638). Also, it is usual to cite Gettelman and Forster (2002) when you refer to the CPT (line 37)(https://doi.org/10.2151/jmsj.80.911). The physical mechanisms mentioned (line 54) are well explained with a model and radiosonde data by Ferreira et al. (2015) (https://doi.org/10.1002/qj.2697). It provides an excellent discussion of some of the most relevant constraints, and it would be worthy of citing it to let the reader get some insight on them.

In lines 57-61, the authors discuss the limitations of satellite data. Indeed, they use AURA-MLS for comparison purposes here. I think that they should cite the works validating WV profiles of AURA-MLS for the SPARC Data Initiative, as they provide the background on the validity and limitations of the measurements. At least one of the coauthors of this manuscript is also coauthor of such works:

Toohey et al. (2013) https://doi.org/10.1002/jgrd.50874 Hegglin and Tegtmeier (2017) The SPARC Data Initiative: Assessment of stratospheric trace gas and aerosol climatologies from satellite limb sounders. SPARC Report No. 8, WCRP-05/2017.

**Author Response (A.2.1)**

We agree with the reviewer and have revised the introduction to be more thorough in the background work.  However, not every work recommended by the reviewer are mentioned. To include all of them, the discussion may become too diffusive.  The introduction has been updated in order to provide better context for our case study and ensure the reader has a broader picture of the background and field. This includes edits to the text that incorporate some of the suggested references, as well as, several additional references that we feel helped to contextualize the discussion.  These changes have significantly improved the introduction and help to set up the overall goal of the paper.

Related to the Introduction: the manuscript has two parts, the validation of the instrument and the case study. Therefore, I think that all the information relevant for the case study that lets to interpret the results should be presented first, included in section 1.

In this vein, the current section 6 should be moved earlier in the manuscript, before beginning the analysis and interpretation of the results. Also, the current figure 1 is right; still, I think that it would be good to include a similar isobaric synoptic map (to check the meteorological situation) and the corresponding map for the first lapse rate tropopause. Doing it would let the reader have a broad picture of the situation. Double tropopauses can happen because of several different conditions, and a priori all of them should be had into account. To do it, all this information is relevant. After it, I suggest to include a brief sentence discussing how the region chosen for the ER-2 flight is one of the central

global areas of occurrence of double tropopauses including summer as Añel et al. (2008) shows (https://doi.org/10.1029/2007JD009697). The other work cited in the text and typical about the study of double tropopauses, Randel et al. (2007) do not show them for July over the region studied in this work.

**Author Response (A.2.2)**

We have made structural changes to the paper and further clarified that the objective of the case study is to identify the process of transport revealed by the observation and that the observation further demonstrate the scientific significance of the new measurement capability.

Specific changes:

1) Firstly, we have included a new figure showing 3 - 48-hour time steps (each day at 20:00 UTC) of PV on the 380 K surface for the 6 days leading up to the date of the case study (this is now Figure 2 in the paper). The Figure clearly shows a Rossby wave-breaking event has occurred in the days preceding the flight that results in mixing along the subtropical jet.

2) We have added a new figure (Figure 3) shows the PV on the 380 K (Figure 3 (a)) and 400 K (Figure 3 (b)) surfaces for the 07/21/2017 18:00 UTC time step. In the Figures, the tropospheric and stratospheric air masses are separated by the 6PVU contour on the 380 K surface and 8 pvu on the 400 K surface. Here it is observed that the mixing associated with the Rossby wave breaking results in a long low PV "tongue" consistent with tropospheric air that extends from the Western Pacific and tracks the subtropical jet across North America.

3) To characterize the vertical structure we have included a Figure (now Figure 4) that shows the height of the thermal tropopause and the location/extent and height of the secondary tropopause for the 07/21/2017 18:00 UTC time step. In these figures one can clearly see that there are several double tropopause regions located on the poleward of the subtropical jet. The SHOW measurements track crosses one of these regions. While additional time steps are not shown, it is useful to point out that the regions of double tropopause vary in extent from time-step to time-step. In fact, the double tropopause region that SHOW crosses becomes larger near the 21:00 UTC time step. A paragraph has been included in the text that discusses this Figure. We believe that the updated analysis provides the relevant context for the case study and justifies the suggestion that the moist filament observed along the second tropopause in Figure 6 (a) is likely of tropospheric origin.

4) The goal here is not to validate the SHOW measurements using the reanalysis data but rather show that the measurements are consistent with a mixing event. Therefore, the current section 6 has been removed and the discussion of the spatial extent of the event is now examined in Section 3 with the new Figures showing the full synoptic picture.

In Figure 2, I had to realise that the values of the horizontal axis are different for each subplot. Right now, it is harder to visualise the latitudinal variation and the assessment of the vertical 'peaks', but it is necessary to be able to compare all of them adequately. Therefore, please, use the same axis for every

subplot. Also, it would be helpful to contextualise the air masses if you can add a horizontal line at the level of the thermal tropopause (first and second, if possible). In the caption, you have missed the degree symbol before the cardinal points.

**Author Response (A.2.3)**

The axes have been adjusted to be the same and we have included horizontal lines noting the altitudes of the first and second tropopause. The degree symbol has been added before the cardinal points in the caption as well as throughout the manuscript where it was missed.

Regarding Figure 3, I have several concerns that should be clarified and better discussed:

First of all, it would be useful in you can include the longitude value in the caption. Secondly, the authors do not say how they have computed the thermal tropopause. Did they use its definition (WMO, 1957)?. Was it retrieved from reanalysis?. It has to be clarified. Also, the use of model levels for the reanalysis could have improved the discussion. If you can use them, it would be better.

**Author Response (A.2.4)**

1. The longitude stayed nearly constant during the flight as mentioned in the text describing Figure 3 and in Section 2. We have added the following line to the caption of the figure: "The longitude is along the 124.5° W line and is nearly constant for the measurements."
2. There is no tropopause product in the ERA-5 reanalysis available to this work. The tropopause we used is derived from the 37 level temperature product. We stated this in the revision and have included a detailed description:

   "Here the tropopause is derived using the ERA-5 temperature field using the lapse rate definition (WMO, 1957; 1992) with a modification. The modified version locates the first tropopause as the lowest level where the lapse rate drops below 2 K/km and remains below that value on average for 1 km (instead of 2 km). A second tropopause is identified if the lapse rate increases above 2K/ km (instead of 3 K/km) and then decreases again below 2 K/km. This is done to remedy the coarse vertical resolution of the of the temperature data. This type of modification has been recognized to allow identification of the double tropopause derived from coarse resolution temperature data that is more consistent with high resolution observational data (Randel et al., 2007). In particular, our goal here is to highlight the spatial extent of the layered static stability structure as discussed in Sections 4-5."

About the plots: in Fig. 3a the isentropic entrainment in the lowermost stratosphere reaches 40.5 degrees N. However, in this region, the PV values are large (up to 6 PVUs, at least 5 PVUs). No doubt, the WV is of tropospheric origin, but such PV values are much higher than acceptable for tropospheric air. At these latitudes, the larger values expected for tropospheric air are 3.5-4 PVUs. If you check your Fig. 4a it seems clear that the 6 PVUs value that you mention in the text as a value for the tropospheric air, is

seen in AURA, not so much in the SHOW measurements (and the AURA measurement fits better with the shape of the potential temperature lapse rate in Fig. 3b). Wang and Polvani (2011) (https://doi.org/10.1029/2010JD015118) and Añel et al. (2012) (http://dx.doi.org/10.1100/2012/191028) have already shown with idealized experiments and lagrangian models how the equatorward movement of air masses trough tropopause breaks at midlatitudes is also very important (and indeed they did it for regions close to the one studied here). Checking the Figure 4b, it could be argued that there is a fingerprint of the movement of stratospheric air equatorward through the break, because of the higher values of ozone that reach the 4 PVUs (near to the more accepted tropopause value) and 36.5 degrees N. Therefore, I think that you need to write the paragraphs from line 204-216 with a more complete discussion and better balance.

**Author Response (A.2.5)**

1) The PV value for representing the tropopause is the topic of Kunz et al., 2011. There the gradient based analysis showed that at 380K the average PV for identifying tropospheric to stratospheric change is 6 pvu. Although not shown it is around 8 pvu at 400K . We include this reference in the revision. We also emphasized in the discussion not the specific PV contour but the weakening of the PV gradient in the region indicates the tropopause break.
2) Yes the discussion focused on the poleward RWB as indicated in the new figures 2-4. For the purpose of this study, the resulting vertical layered structure above the subtropical break is the key. Equator-ward transport is also important but not the focus of this study.

A right way of checking the reality of the movement would be with using a lagrangian transport model (as in Añel et al. 2012). If you can include it, it would be a great addition to the manuscript, but I realise that it is not the goal of this work, so I do not consider it a 'must' here. But given that you do not provide it and on the ground of the tasks that I mention above, along with the text, you should relax the language and the level of the statement about poleward isentropic mixing. For example, in line 197, where you say 'it is widely accepted' because there are many different synoptic situations.

**Author Response (A.2.6)**

Indeed we focus the analysis on providing the high resolution measurement and evidence of the transport impact on lowermost stratospheric water vapor and the highlight of new measurement capability. This is clarified in the opening and abstract.

Finally, the summary and conclusion' section is short. I would include some discussion on the error of SHOW and how it could have impacted the results here presented. Also, I would find it interesting to include a reflection on the limitations of SHOW to sample similar episodes in more poleward latitudes. Given that SHOW seems to have a restriction below 13.5 km and that poleward the tropopause height decreases, is SHOW limited to sample these episodes only around the subtropical jet?. In line 304, the terminology of 'tropospheric intrusions' is used again. As said before, I think that talking about 'tropopause breaks' is correct in the context of this work.

**Author Response (A.2.7)**

We have updated Section 6 to be an expanded discussion and conclusion section. This section provides discussion on the limitations of the SHOW measurements and how these limitations may have impacted the study is now included in the paper. Specifically, we discuss the choice of the lower altitude cutoff. The lowest altitude cutoff of the measurements is primarily associated the optical depth. At some point, when the optical depth is below 1, scattered light from below is fully absorbed by the atmosphere. This generally occurs a few 3-5 km below the tropopause and varies from profile to profile. Algorithms are in development to actively determine this cutoff during the retrieval process. However, we did not have apriori knowledge of the meteorological picture prior to performing the retrievals. For the current study we chose 13.5 km to fix the altitude at a reasonable height (several km below the expected 15 km -18 km tropopause height for latitudes below the break) that we knew would provide accurate retrievals across the latitude range.

We also reiterate in the discussion section that we are not trying to validate the SHOW measurements with MLS and the reanalysis data. The SHOW measurements provide a much higher spatial sampling compared to either MLS or ERA5 and we are confident in the quoted uncertainties of the SHOW measurements. Therefore, the variability observed in the two-dimensional water vapour distribution observed by SHOW is representative of the true state of the atmosphere. The reanalysis data provides the appropriate meteorological context and the MLS measurements serve to geophysical consistency with the SHOW measurements.

**Minor corrections (A.2.8)**

References: - The list of references is in the incorrect order. - Randel et al. 2007a and 2007b are not distinguished in the list of references.

- Only Randel et al., 2007a is referenced the paper now

Line 28 - de Forster and Shine Line 30 - Gettelman and Sobel, 2000 Line 44 - Appenzeller and Davies, 1992 Line 52 - Pan et al. 2010 is not listed among the references

- Gettelman and Sobel, 2000 Line 44 - Appenzeller and Davies, 1992 Line 52 - Pan et al. 2010 are no longer referenced in the manuscript
- de Forster and Shine, 1999 and 2000 are included

Table1-unitsofspeed–km/h(notkm/hr)

- Corrected

Line112-the international units of pressure are hPa, not mb

- Corrected

Line 133 - coarser

- Corrected

Line 168 - there is not an orange line in the figure

- Corrected

Lines 176-178 - this sentence is redundant. It has been discussed earlier in the text. I suggest removing it.

- Corrected

Lines 187-189 - in line 188 when you refer to the tropopause, it is hard to know if you refer to the first or the second; please clarify it. Also, it would be useful in you can include in the plot something (an 'A' and a 'B,' a star and a square,...) to make clearer to what region you refer.

- The text has been edited to be less ambiguous

Lines 248-251: the last sentence is obvious and can undermine the achievements of the SHOW instrument. I recommend to move it to the conclusions as a final reflection.

- This statement has been moved to the new discussion section (Section 6)

Line 269 - 20 degrees Figure 5 - caption - I would say '230 and 235.5'. I read the plots from top to bottom and the one on the top corresponds to 230. As it is now, it can be not very clear.

- Corrected

---

## Author Response (AR2)

**Review of 'Observational evidence of moistening the lowermost stratosphere via isentropic mixing across the subtropical jet' by Langille et al. submitted to Atm. Chem. Phys. (acp-2019-220)**

The authors have submitted a reviewed version of their manuscript, much improved and better focused. I think that it can be published after a few minor revisions.

**Author response**: Thank you for your suggestions and comments. Please find responses to each minor correction in blue below. The marked up manuscript can be found attached as well.

Minor corrections:

**L35:** Please, include a reference to the description of the SHOW instrument

A reference to Langille et al., 2018 has been included.

**L37-41**: It is correct to mention here that the goal is to study the event; however, most of this text is a statement on the results that should be moved to the 'Discussions and Conclusions'

We do not believe that the text should be moved to the Discussion and Conclusion section. The statement that we have included clarifies the point of the paper and the importance of the results to the reader. We feel that this is important to include in the introduction.

**L43-38**: Respectfully, the text in these lines shows careless exposition and writing. No doubt, with 'drop' you want to say that the extratropical tropopause is at an altitude lower than the tropical one. However, saying that it 'drops' is not of a proper explanation about climatological characteristics and the physical mechanisms that drive to such behaviour (that have nothing to do with a 'drop')

The word "drop" may be a bit too colloquial. Therefore, we replace the word "drop" and edit the line to say: "A ubiquitous feature here is a sudden decrease in the altitude of the thermal tropopause".

L51-52: As it is written right now, it could be understood that wave breaking is the only mechanism associated with it. Please, modify the text to make clear that it is part of the existing possibilities "...is associated (among others) with Rossby-wave breaking and large-scale poleward transport." For example, a nice addition to frame the topic here could be to mention that the increase of vertical baroclinicity is also associated with double-tropopauses. This phenomenon was observed by Castanheira et al. (https://doi.org/10.5194/acp-9-9143-2009) using normal modes (therefore filtering Rossby waves), and it is a well-known impact of global climate change.

The text is meant to focus the reader on the fact that the lowermost atmosphere is strongly influenced by isentropic mixing associated with Rossby wave breaking and that the double tropopause "can" be associated with it. The current wording does not suggest that wave breaking is the only mechanism associated with it. In any case, in order to ensure the text is clear, the line in question has been edited to read: "A number of more recent studies have shown that the occurrence of a double tropopause **can be** associated with Rossby- wave breaking and large-scale poleward transport." The inclusion of "**can be**" here ensures it is clear that we are not suggesting it is the only mechanism associated with it. Including a broader discussion of other mechanisms responsible for the generation of multiple tropopauses would distract the reader from the point of the paper.

**L52**: signature?

"Signatures" is used here instead of "signature" since there is indeed more than one type of variability or "signature" imprinted on the spatial distribution of the trace species that can be associated to wave breaking.

**L56-65:** most of this information is repeated later in Section 2. I suggest to include here only a simple comment on the use of SHOW and refer the reader to such section.

We have removed the quoted lines below since it is repeated in Section 2 and referred the reader to Section 2 for more details on the instrument specifications:

"The instrument implements a limb imaging spatial heterodyne spectrometer (SHS) to obtain vertically resolved images of the water vapour spectrum using limb-scattered sunlight in a 2 nm spectral window centered on 1364.5 nm (Langille et al., 2017). Each SHOW measurement is inverted using the optimal estimation approach to obtain the vertical water vapour profile for each along-track sample (Langille et al., 2018)."

L72: Absolute values are not too informative. I understand that this refers to previous work, but if possible, I would suggest adding relative errors or percentages of bias compared to the values measured by the radiosondes.

A slight positive bias of 3.3% was recorded between the sonde measurements and SHOW presented in early paper. Remaining percent differences are found to be from +/- 10%. Differences between them are expected due to difference in the observed column of air, viewing geometry from SHOW, measurement uncertainty and known issues with the accuracy of the radiosonde at these altitudes. However, as discussed in Langille et al.(2019) differences between the radiosonde and SHOW can be (and were) used to check consistency and general shape (and magnitude) of the profile between the two measurements. The text has been edited to report the %bias and %difference between the measurements as shown below:

**"….with the radiosonde recording a positive bias of ~ 3.3% relative to SHOW and percent differences of < ±10 % , due to both natural variability between the observations and measurement precision."**

L85-90: I support the view of the authors of avoiding the inclusion of unnecessary discussions in the Introduction. I prefer it too. However, this paper is not so long to consider it unreadable. It is seventeen pages long in its current form, including abstract, twenty-nine references (three pages) and another seven pages for figures. This results in roughly six pages for the Introduction, Methods, Results and Discussion. That said, I do not find a good reason to avoid including relevant information that is necessary from the formal point of view. For example, the paper does not have a 'Data availability' section (mandatory in many journals) and in the lines here referred there is no information about the source for the ERA-5 or AURA-MLS data. It must be said which is the source for the data: Is it the ERA5 repository in the ECMWF? A local copy at the University of Saskatchewan or NCAR? Was the dataset retrieved through the Internet? If yes, when was it last accessed?.

Also, it would be desirable to include a Zenodo repository with the data files containing the SHOW measurements used in this work. Do not get me wrong; this information is necessary to assure independent reproducibility of the work. Therefore, I recommend the authors to take into account at least some of these recommendations. They will improve the manuscript.

We have added the following Data Availability section at the end of the paper:

*Data Availability. The ERA-5 reanalysis product was downloaded from the ECMWF online repository which can be accessed at https://cds.climate.copernicus.eu/cdsapp#!/dataset/reanalysis-era5-pressure-levels. The AURA-MLS version v4.2 data was downloaded from the GES-DISC link found at https://ml.jpl.nasa.gov/data/. The SHOW data is available upon request from the author.*

**L92**: ...is a spatial heterodyne

Corrected in the text

**L115**: Langille et al. (2019)

Corrected in the text

**L121**: 1 hPa to 1000 hPa

Corrected in the text

L133: Kunz et al. say that the typical PV values for the tropopause range between 1.5 and 5 PVU and in Fig. 3 the transition values highlighted are 6 and 8 PVU. Moreover, it seems hard that the transition is located along such isolines. I understand that the authors have not performed specific computations for the corresponding tropopause-PV values in this case, Right? This should be acknowledged in the text, saying that the used values are an informed guess.

**Response to this comment:**

One of the main findings of Kunz et al., 2011 is that based on the isentropic gradient, the PV value representing the tropopause increases with the isentropic levels. In Figure 6 of the paper, this point is shown quantitatively (see this Figure reproduced below). For the JJA season, the average tropopause PV value for the 380K is greater than 6 PVU.

This point is verified and supported by trace gas measurements-based PV tropopause. One example is shown in Kunz et al., 2011b (Figure 7 included below)

Figure 6 from Kunz et al., 2011a:

[Figure]

**Figure 6.** Zonal mean PV in PVU with equivalent latitude for different seasons in 2002. The dynamical tropopause, $\varphi_e^{TP}$ (yellow diamonds), and the transition region, $\varphi_e^{B}$ (open yellow diamonds), are shown on each isentrope. The height of the thermal tropopause, $\theta_{TP_e}$, is shown by white dots. Specific PV isolines between ±1 and ±6 PVU are highlighted by black solid lines. The 2 PVU isoline is a black dashed line and the zonal wind is shown in red.

Figure 7 from Kunz et al., 2011b:

[Figure]

**Figure 7.** WACCM CO distribution at 380 K on (a) 14 April 2008 and (b) 15 April 2008. $PV^{TP}$ is represented by the 6.1 PVU isoline on 14 April and by the 6.0 PVU isoline on 15 April 2008 (yellow line). The 4 PVU isoline (black line) and the horizontal wind speed (orange contours) are also shown.

There has not been a systematic study of PV horizontal gradient at the 400 K level. The use of 8 pvu is consistent with the increasing tendency revealed in Kunz et al. 2011 and also consistent with the observed dynamical structure.

To be more accurate without over burden the statement, we made the following revision to this sentence:

*"Here, we used 6 pvu to identify the separation between tropospheric air on the 380 K surface (Kunz et al., 2011), which is noted by the white transition region between red (low PV air and tropospheric) and blue (high PV air and stratospheric) colors in the figures. Similarly, we used 8 pvu to represent this separation on the 400 K isentropic surface, which consistently highlighted the filament of tropical air (more tropospheric) in the background of extratropical (lower stratospheric) background."*

**References:** (the first one is already in the paper. The second one is just for the referee or editor to see)

Kunz, A., P. Konopka, R. Müller, and L. L. Pan (2011a), Dynamical tropopause based on isentropic potential vorticity gradients, J. Geophys. Res., 116, D01110, doi:10.1029/2010JD014343.

Kunz, A., L.L. Pan, P. Konopka, D. E. Kinnison, and S. Tilmes (2011b), Chemical and dynamical discontinuity at the extratropical tropopause based on START08 and WACCM analyses, J. Geophys. Res., 116, D24302, doi:10.1029/2011JD016686.

**L141:** The WMO (1992) reference is not in the list.

Corrected in the text.

**Figure 5**. In the caption, 'SHOW' should be capitalized. Also, it should not appear a blank space between the degree symbol and the letter for the cardinal point.

Corrected in the text.

**Figure 6**. In the caption 'light grey' corresponds to PV, and zonal wind is in black.

Corrected in the text.

**L244**: HIRDLS

Corrected in the text

**L251**: PVU

Corrected in the text

**L261**: ' ...which is a sign of irreversible transport.'

Corrected in the text

**Figures 6B and 7C**: You use ERA5, so change ECMWF by ERA5 in the titles. The ECMWF has many reanalysis products, and as it is now, it is not clear enough what you mean. Readers could find it confusing.

Corrected in these figures

**L327**: Langille et al. (2018)

Corrected in the text

**L340**: 2018)

Corrected in the text

The citation style of this journal uses parenthesis, not brackets.

Corrected in the text

**Additional author corrections:**

**Figure 6b. Figure 7b, Figure 7c:**  We adjusted the saturation limits so that values that are off the scale are no longer white. In Figure 7b the lower limit was also adjusted to bring out the spatial structure.

**Figure 7a:** The altitudes in the original Figure were calculated directly from the MLS pressure levels assuming a fixed scale height and reference pressure. The Figure has been updated with calculated altitudes from the MLS pressure levels using the relationship between the ERA5 pressure levels and the geometric height calculated from the ERA5 geopotential height. The calculation is more accurate and results in only minor changes to the Figure.

**Minor corrections suggested by the Editor:**

P7, L183: within in -> either "within" or "in"

Corrected in the text

P8, Figure 5 caption: colon after Figure 5 missing (this holds also for the other figures).

Corrected in the text

P8, FIgure 5 caption: show -> SHOW?

Corrected in the text

P10, L239: sits -> is located

Corrected in the text

P10, L241: has layered structure -> has a layered structure

Corrected in the text

P10, L244: HIRDLES -> HIRDLS

Corrected in the text

P10, L251: pvu -> PVU

Corrected in the text

P11, L286: degree sign is missing.

Corrected in the text

[revised manuscript text omitted]